# Network analysis reveals abnormal functional brain circuitry in anxious dogs

Yangfeng Xu[1,2‡]*, Emma Christiaen[3‡], Sara De Witte[1], Qinyuan Chen[1], Kathelijne Peremans[2], Jimmy H. Saunders[2], Christian Vanhove[3], Chris Baeken[1,4,5]

**1** Ghent Experimental Psychiatry (GHEP) Lab, Department of Head and Skin, Faculty of Medicine and Health Sciences, Ghent University, Ghent, Belgium, **2** Department of Morphology, Imaging, Orthopedics, Rehabilitation and Nutrition, Faculty of Veterinary Medicine, Ghent University, Ghent, Belgium, **3** Medical Image and Signal Processing (MEDISIP), Department of Electronics and Information Systems, Faculty of Engineering and Architecture, Ghent University, Ghent, Belgium, **4** Department of Psychiatry, Faculty of Medicine and Pharmacy, Vrije University Brussels, Brussels, Belgium, **5** Department of Electrical Engineering, Eindhoven University of Technology, Eindhoven, The Netherlands

‡ YX and EC have contributed equally to this work and share first authorship.
* Yangfeng.Xu@UGent.be

**Data Availability Statement:** The dataset is available on request from the GISMO (the Integrated Research Information System of Ghent University) via gismo@ugent.be, as the dog MRI

## Abstract

Anxiety is a common disease within human psychiatric disorders and has also been described as a frequently neuropsychiatric problem in dogs. Human neuroimaging studies showed abnormal functional brain networks might be involved in anxiety. In this study, we expected similar changes in network topology are also present in dogs. We performed resting-state functional MRI on 25 healthy dogs and 13 patients. The generic Canine Behavioral Assessment & Research Questionnaire was used to evaluate anxiety symptoms. We constructed functional brain networks and used graph theory to compare the differences between two groups. No significant differences in global network topology were found. However, focusing on the anxiety circuit, global efficiency and local efficiency were significantly higher, and characteristic path length was significantly lower in the amygdala in patients. We detected higher connectivity between amygdala-hippocampus, amygdala-mesencephalon, amygdala-thalamus, frontal lobe-hippocampus, frontal lobe-thalamus, and hippocampus-thalamus, all part of the anxiety circuit. Moreover, correlations between network metrics and anxiety symptoms were significant. Altered network measures in the amygdala were correlated with stranger-directed fear and excitability; altered degree in the hippocampus was related to attachment/attention seeking, trainability, and touch sensitivity; abnormal frontal lobe function was related to chasing and familiar dog aggression; attachment/attention seeking was correlated with functional connectivity between amygdala-hippocampus and amygdala-thalamus; familiar dog aggression was related to global network topology change. These findings may shed light on the aberrant topological organization of functional brain networks underlying anxiety in dogs.

data is the property of Ghent University. A research agreement will be arranged for data share.

**Funding:** This study is funded by Belgium governmental FWO institution (Project number G011018N). Emma Christiaen is an SB PhD fellow at Research Foundation - Flanders (Project number 1S90218N) The funders had no role in study design, data collection and analysis, decision to publish, or preparation of the manuscript.

**Competing interests:** The authors have declared that no competing interests exist.

## Introduction

Anxiety disorders include disorders that share features of excessive fear and anxiety and related behavioral disturbances [1]. Anxiety disorders are classified as social anxiety disorder (SAD), post-traumatic stress disorder (PTSD), generalized anxiety disorder (GAD), panic disorder, and specific phobias [2]. These disabling conditions cause significant burdens to the individual and society, such as causing social relationships, suicide, and increasing healthcare costs. It has been widely reported that characteristic alterations in structural and functional connectivity (FC) are associated with anxiety [3–5], although the functional integrity and topological organization in such patients remains largely unclear.

Animal models are indispensable tools to unravel neurobiological mechanisms underlying anxiety disorders and their pathological variations. Change in neuronal activities in specific brain areas correlated with anxiety have been reported in primates [6], rodents [7], and dogs with pathological anxiety [8]. The investigation of the canine species could be of particular interest. It has been well accepted that dogs can be valid translational models for a number of human behavioral disorders [9]. Dogs can also develop these mental illnesses, and they are also relatively easily accessible and manageable compared to primates. Moreover, compared to rodents, dogs have a larger amount of frontal cortex. Thus, the canine species might be an appropriate model to investigate brain networks involved in anxiety, and together with other animal research, such as rodents, can be used as a model for human anxiety (and vice versa). The prevalence of anxiety disorders in the dog is high and the most encountered behavioral disorder in daily practice [10]. Moreover, they form a serious welfare problem not only for the well-being of the individual, but they also compromise the relationship with the owner leading to abandonment, rehoming, or even euthanasia. In the case of comorbid aggression, they result in safety hazards and are of public concern. It has been demonstrated that in several canine neuropsychiatric disorders, the neurobiological base has similar characteristics as its human counterparts [11–13], also in dogs [14]. However, till now there is no report about rs-fMRI studies in anxious dogs, even though their emotional value to humans puts an increased demand on veterinarians to implement refined diagnostic tools and provide optimized treatment. Thus, we hypothesized that similar abnormal regional neural connectivity as in humans diagnosed with anxious behaviors could be found in anxious dogs.

Resting-state functional magnetic resonance imaging (rs-fMRI) could reveal correlated spontaneous low frequency blood oxygenation level-dependent (BOLD) fluctuations in anatomically distinct regions called "resting state networks" (RSNs), which are thought to reflect neural activity or relevant functions that occur in the grey matter [15]. Functionally connected regions can be identified, and brain networks can be detected based on statistical dependencies between the BOLD time series of brain regions of interest. There are several analysis methods available for investigating the brain organization, including seed-based correlation, independent component analysis and graph theory [16]. Graph theory has been widely applied to analyze the topological properties alterations in neuropsychiatric disorders and enabled understanding of how brain disorders affect the brain cognitions based on fundamental properties of the brain network, including anxiety, depression, schizophrenia, Alzheimer's disease, and epilepsy [17]. In graph theory, the brain is considered a network or graph with brain regions as nodes and the relationship between the nodes as edges. The brain network can then be described and quantified using graph theoretical network metrics [18]. Specifically, nodal degree measures the degree of nodes tending to cluster together, global efficiency measures the efficiency of parallel information transfer through the network, clustering coefficient measures the efficiency of information exchange within a local subnetwork or among adjacent regions, characteristic path length measures the ability for information

propagation within the network, the small-world network indicates a typical network that has similar path length but higher clustering than a random network [18–20].

In this study, a combination of rs-fMRI and graph theory was used to investigate the underlying neuronal mechanisms of action of anxiety in dogs. rs-fMRI data were acquired in patient dogs with anxiety and in healthy dogs. In addition, different symptoms of anxiety were assessed using the Canine Behavioral Assessment & Research Questionnaire (C-BARQ), a canine behavioral questionnaire. The aim of this study was threefold: 1) to evaluate differences in brain network topology between healthy dogs and dogs with anxiety; 2) to identify differences in FC in regions implicated in anxiety and 3) to assess whether different symptoms of anxiety, as measured with the C-BARQ, are related to specific functional network differences. The results in dogs will benefit both veterinary medicine for anxiety-disordered animals and may serve human medicine as a natural model.

## Materials & methods

### Animal

Twenty-five beagle dogs were recruited as the healthy group (6 castrated males and 19 neutered females; aged between 1 and 8 years old; Table 1). These dogs were owned by the Department of Small Animals and the Department of Veterinary Medical Imaging and Small Animal Orthopedics, Ghent University. All dogs were checked every 3 months for health monitoring, spontaneous behavior in the kennel and behavioral responses in different contexts. All healthy beagle dogs were housed in groups of eight on an internal surface of 15 m$^2$, with access to an outside area of 15 m$^2$. The floor covering in the inner part consisted of wood shavings. Toys were given to these dogs every day and they were released to an enclosed playground twice a day. In addition, the veterinary students and animal house managers walked the dogs regularly. Furthermore, all these dogs displayed normal behavior, evaluated by both veterinarians involved and care takers of the dogs regularly (Details in S1 Text). Behavior remained impeccable over the whole study period. The patient group consisted of 13 volunteer dogs. The Ghent University Ethical Committee approved this study and all guidelines for animal welfare, imposed by the Ethical Committee, were respected (EC number, 2015–140, 2018–09, 2018–088).

### Patient recruitment

Based on the dog's history, the physical examination and the questionnaires, all patient dogs were diagnosed with anxious behavior, with or without aggression, specifically towards familiar and unfamiliar people and animals. Mean problem duration of these dogs ranged from 8 months to 2 years; 6 were adopted from the shelter, 7 were raised up at home; 5 were aggressive towards people and dogs; 9 were afraid of people and dogs; 6 showed noise phobias; 4 went for behavioral therapy but failed; 3 went for drug therapy but failed. Blood samples were taken for thyroid function tests and sent to a commercial lab (Zoolyx, Aalst, Belgium) to exclude thyroid dysfunction-led behavior problems [21].

**Table 1. The demographic information of the control group (n = 25).**

| Breed | Number | Gender | Age (month, mean ± sd) |
|---|---|---|---|
| Beagle | 19 | FC | 47 ± 23 |
| | 6 | MC | 64 ± 32 |

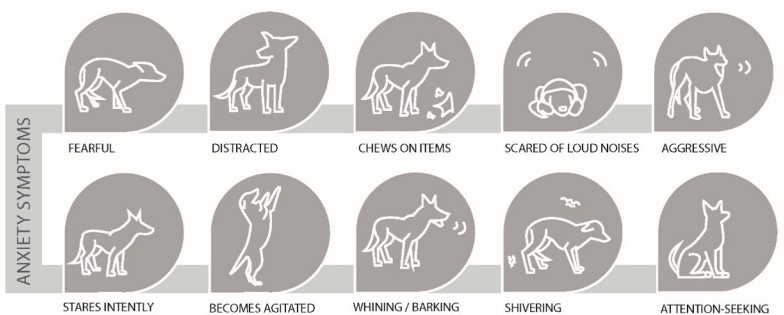

**Fig 1. Anxiety symptoms of dogs.**

## Behavior evaluation

The dog's behavior (Fig 1) was assessed using the validated canine behavioral questionnaire filled in by the owner. The Canine Behavioral Assessment & Research Questionnaire (C-BARQ) is a standardized, behavioral evaluation tool for dog owners/guardians, handlers, and professionals. It was developed and validated by Y Hsu and J Serpell [22, 23]. It provides information concerning the dog's behavior and temperament in 13 scales. This questionnaire contains 101 questions grouped into seven sections: training and obedience, aggression, fear and anxiety, separation-related behavior, excitability, attachment and attention seeking and miscellaneous. The responses to the questions were scored with a 5-point frequency scale or a 5-point semantic differential scale. The questionnaire was translated in Dutch.

A second questionnaire was used, not validated yet, but specifically developed for current research questions at our department (personal communication K Overall [24]). This questionnaire did not include scoring and was predominantly used to get an overview of the different anxious and aggression driven reactions in different situations, including owner information, dog information, separation anxiety and noise phobia/reactivity screen, reactivity and aggression screen, stereotypical (repetitive) and ritual behavior. Like the C-BARQ, the questions were translated in Dutch.

## MRI scan

All neuroimaging data were collected on a 3T Siemens Trio Tim scanner with the phased-array spine coil and a phased-array body matrix coil. The dogs were pre-medicated in a quiet room with dexmedetomidine (Dexdomitor; Orion) at 375 μg/m2 by intramuscular injection. General anesthesia was induced with 2–3 mg/kg propofol (Propovet Multidose, Abbott Laboratories) intravenously through a cephalic vein catheter. Anesthesia was maintained with isoflurane (Isoflo, Abbott Laboratories) in oxygen given to effect.

For each dog, a high-resolution T1-weighted 3D image was firstly acquired using a MP-RAGE sequence with the following parameters: repetition time (TR) = 2250 ms, echo time (TE) = 4.18 ms, matrix size = 256 × 256, field of view (FOV) = 256 × 256 mm2, flip angle = 9˚, and voxel size = 1 × 1 × 1 mm3, 176 slices. Then, a rs-fMRI scan was acquired using a single-shot gradient echo planar imaging (EPI) sequence (TR/TE = 2000/27 ms, flip angle = 90˚, matrix size = 64 × 64, FOV = 192 × 192 mm2, voxel size = 3 × 3 × 3 mm3; slice thickness = 3 mm without inter-slice gap; number of slices = 24). The dogs were placed headfirst and sternally in the scanner bore. After completion of the MRI acquisition, the dogs were allowed to recover.

## Data analysis

**Preprocessing.** The rs-MRI data were preprocessed using SPM12 (https://www.fil.ion.ucl.ac.uk/spm/software/spm12/) and MRtrix3 [25]. First, images were realigned to their mean image using a least squares approach and a 6 parameter (rigid body) spatial transformation to remove motion artifacts. Next, the images were registered to an EPI template and spatially smoothed using a Gaussian kernel with a Full Width at Half Maximum of 6 mm. Afterwards, a band pass filter (0.01 Hz–0.1 Hz) was applied to remove physiological and low frequency noise. During the registration to the EPI template, differences in brain size between the dog breeds were minimized. During the processing, field inhomogeneity-related artifact correction was not performed, global white matter (WM) and cerebrospinal fluid (CSF) were not regressed [26].

**Functional network construction.** A parcellated atlas (adapted based on former research) [27, 28] containing 30 cortical and subcortical regions of interest (ROIs) was constructed based on the T1-weighted anatomical images of all animals using MRtrix3. Based on our former research and others' research, we chose these regions as ROIs [8, 13, 29–34]. One of the major objectives of our canine studies—in relation to similarities (or not) in brain circuits- is for instance to use the canine brain model to improve non-invasive brain stimulation methods, also in humans [35–37]. These ROIs are listed in Table 2 and visualized in Fig 2A. Using a Graph Theoretical Analysis Toolbox (GRETNA) [38], the mean time series of each ROI was extracted and the Pearson correlation coefficient was calculated between each pair of ROIs. For each rs-fMRI scan, a 30 x 30 correlation matrix was obtained. Only positive weights in brain connectomes were included, and the network measures for all the nodes were computed on the weighted networks. To remove the weakest connections, thresholds based on network density (i.e., the number of remaining connections divided by the maximum number of possible connections) were applied to the correlation matrices. Thresholds were chosen to obtain network densities ranging from 20% to 50%, with a 5% interval [18, 39]. After thresholding, the correlation coefficients were Fisher r-to-z transformed to obtain a normal distribution. Then, functional networks or graphs were constructed based on the correlation matrices. In these graphs, the nodes correspond with the ROIs and the edges with the correlation coefficients [20, 38].

**Graph theoretical analysis.** For all graphs, several network metrics were calculated using GRETNA, both on a global (entire network) and nodal (per ROI) level. Degree or connection strength is the number of (weighted) edges connected to a node. It is a measurement of centrality and indicates how important a node is in the network. The characteristic path length (Lp) is the average number of edges connecting two nodes in the network and global efficiency (Eglob) is the average inverse path length between two nodes. These are measures of functional integration or overall communication efficiency in the network. Clustering coefficient (Cp) is the ratio of neighbors of a node that are connected to one another as well and local efficiency (Eloc) is the average inverse path length within the neighborhood of a node, i.e., the nodes connected to that node. These are measures of functional segregation or local interconnectivity [18, 40]. Small-worldness (σ) indicates a typical network that has similar path length but higher clustering than a random network [19]. Network metrics were calculated at different correlation matrix densities, from 20% to 50% density with a 5% interval, and averaged over these densities [20, 38]. The pipeline of the data analysis is shown in Fig 2B. Based on our former research [8, 13, 29, 30, 41, 42], relevant veterinary behavior medicine research [43], and human medicine [31–34], we choose these five regions of interest as the anxiety circuit in this study: amygdala, frontal lobe, hippocampus, mesencephalon and thalamus. The vermis was chosen as the control region. Therefore, between-group differences in the nodal level were

**Table 2. Brain regions included in the parcellated atlas.**

| Label | Region |
|---|---|
| 1 | Temporal lobe L |
| 2 | Parietal lobe L |
| 3 | Occipital lobe L |
| 4 | Frontal lobe L |
| 5 | Anterior cingulate gyrus L |
| 6 | Posterior cingulate gyrus L |
| 7 | Hippocampus L |
| 8 | Thalamus L |
| 9 | Caudate nucleus L |
| 10 | Piriform lobe L |
| 11 | Insular cortex L |
| 12 | Amygdala L |
| 13 | Cerebral hemisphere L |
| 14 | Vermis L |
| 15 | Temporal lobe R |
| 16 | Parietal lobe R |
| 17 | Occipital lobe R |
| 18 | Frontal lobe R |
| 19 | Anterior cingulate gyrus R |
| 20 | Posterior cingulate gyrus R |
| 21 | Hippocampus R |
| 22 | Thalamus R |
| 23 | Caudate nucleus R |
| 24 | Piriform lobe R |
| 25 | Insular cortex R |
| 26 | Amygdala R |
| 27 | Cerebral hemisphere R |
| 28 | Vermis R |
| 29 | Mesencephalon |
| 30 | Diencephalon |

Note: L, left hemisphere; R, right hemisphere.

assessed in the regions of the anxiety circuit and vermis. The nonparametric Mann-Whitney U test was performed in the nodal level analysis to compare the between-group differences. To correct the network parameters for multiple comparisons, false discovery rate (FDR) correction was applied by using p < .05 as the significant threshold. The calculation of network measures of integration, segregation and centrality is summarized in Fig 3 [18].

**Statistical analysis.** Differences in the functional brain network were assessed on three levels. On a 1) global and 2) nodal level, group differences in degree, characteristic path length, global efficiency, clustering coefficient and local efficiency were assessed using a Mann-Whitney U test with a significance level of 0.05 after FDR correction for correcting multiple comparisons between nodes. On a 3) connection level, group differences in z-values between regions of the anxiety circuit were assessed using a Mann-Whitney U test with a significance level of 0.05 after FDR correction for correcting multiple comparisons between connections.

The nonparametric Spearman's rank correlation coefficient between global network metrics, nodal degree, global efficiency as a measure for integration and clustering coefficient as a

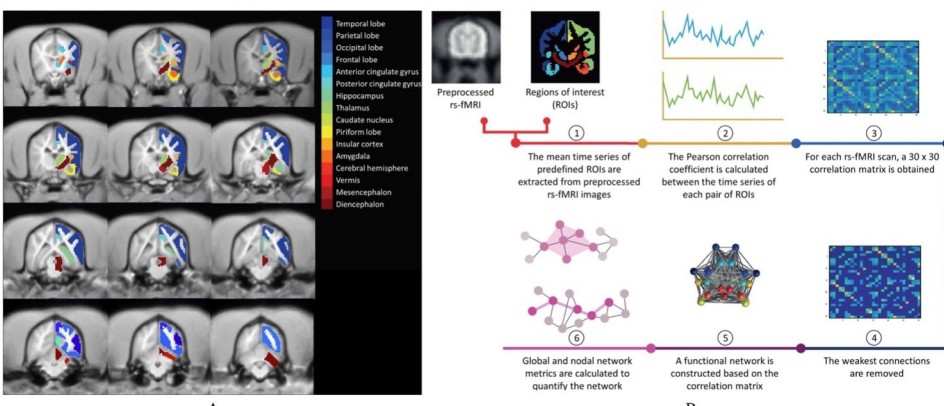

**Fig 2.** A. Regions of interest (ROIs) overlaid on a T1 template. Only the right ROIs are visualized. The left ROIs are identical. B. Pipeline of the data analysis: 1) the mean time series of predefined ROIs are extracted from the preprocessed rs-fMRI images, 2) the Pearson correlation coefficient is calculated between the time series of each pair of ROIs, 3) for each rs-fMRI scan, a 30 x 30 correlation matrix is obtained, 4) the weakest connections are removed, 5) a functional network or graph, in which the nodes correspond with the ROIs and the edges with the correlation coefficients, is constructed based on the correlation matrix, and 6) global and nodal network metrics are calculated to quantify the network.

### Calculation of network measures

**Integration**

Distance: shortest path between nodes i and j
$$d_{ij} = \sum_{a_{uv} \in g_{i \leftrightarrow j}} f(w_{uv})$$
f: map from weight to length (e.g. inverse)

Characteristic path length
$$Lp = \frac{1}{n} \sum_{i \in N} \frac{\sum_{j \in N, j \neq i} d_{ij}}{n-1}$$

Global efficiency
$$Eg = \frac{1}{n} \sum_{i \in N} \frac{\sum_{j \in N, j \neq i} (d_{ij})^{-1}}{n-1}$$

**Segregation**

Number of triangles around node i
$$t_i = \frac{1}{2} \sum_{j,h \in N} (w_{ij} w_{ih} w_{jh})^{1/3}$$

Clustering coefficient
$$Cp = \frac{1}{n} \sum_{i \in N} \frac{2t_i}{k_i(k_i-1)}$$

Local efficiency
$$Eloc = \frac{1}{2} \sum_{i \in N} \frac{\sum_{j,h \in N, j \neq i} (w_{ij} w_{ih} [d_{jh}(N_i)]^{-1})^{1/3}}{k_i(k_i-1)}$$
$d_{jh}(N_i)$: length of the shortest path between j and h that contains only neighbors of i

Modularity
$$Q = \frac{1}{l} \sum_{i,j \in N} [w_{ij} - \frac{k_i k_j}{l}] \delta_{m_i, m_j}$$
$l = \sum_{i,j \in N} w_{ij}$: sum of all weights, $\delta_{m_i,m_j}=1$ if $m_i=m_j$ and 0 otherwise

Small-worldness
$$S = \frac{Cp/Cp_{rand}}{Lp/Lp_{rand}}$$

**Centrality**

Degree
$$k_i = \sum_{j \in N} w_{ij}$$

Participation coefficient
$$y_i = 1 - \sum_{m \in M} \left(\frac{k_i(m)}{k_i}\right)^2$$
$k_i(m)$: number of links between i and all nodes in module m

Betweenness
$$b_i = \frac{1}{(n-1)(n-2)} \sum_{h,j \in N, h \neq j, h \neq i, j \neq i} \frac{\rho_{hj}(i)}{\rho_{hj}}$$
$\rho_{hj}$: the number of shortest paths between h and j

**Fig 3. Formulas to calculate the network measures.**

measure for segregation, and connection strength on the one hand, and the severity of anxiety symptoms, derived from the C-BARQ questionnaire, on the other hand, was calculated to assess whether they were related. Correlations between connection strength and anxiety symptoms were corrected for multiple comparisons using the FDR at q = 0.05.

## Results

### Thyroid test results

All patients' thyroid test results were normal. The total thyroxine (T4), thyroid stimulating hormone (TSH), free triiodothyronine (fT3) and free thyroxine (fT4) were in normal ranges. Thyroglobulin test was negative in all patients (Table 3).

### Global network topology

Five network metrics were calculated on a global level: 1) mean degree, 2) characteristic path length, 3) global efficiency, 4) clustering coefficient, 5) local efficiency and 6) small-world (Fig 4A). No significant differences between groups could be found in these metrics using the Mann-Whitney U test. In addition, we found $\sigma > 1$ for the two groups, indicating that both patient and control groups exhibited small-world attributes.

### Nodal degree in the anxiety circuit

Differences in nodal degree, characteristic path length, global efficiency, clustering coefficient and local efficiency were assessed in the regions of the anxiety circuit: amygdala, frontal lobe, hippocampus, mesencephalon, and thalamus (Fig 4B). Network metrics were averaged between the left and right components of each region. In the amygdala, we found that the patient group exhibited significantly increased global efficiency and local efficiency (p = 0.007, FDR correction; p = 0.003, FDR correction, respectively), and significantly decreased characteristic path length (p = 0.006, FDR correction), compared with the control group (the network measure consistency was checked across the range of thresholds, results are provided in S1 Fig).

### Individual connections in the anxiety circuit

To assess altered connection strength in the anxiety circuit, the z-values of the correlation coefficients between regions of the anxiety circuit were investigated. The z-values are the average of the intra- and interhemispheric connections between the left of right components of brain regions. If a z-value is zero, this means the correlation is below the threshold of an average of 20% to 50% network density [38]. The z-values or connection strength are significantly higher in the patient group compared to the control group for the connections amygdala-hippocampus, amygdala-mesencephalon, amygdala-thalamus, frontal lobe-hippocampus, frontal lobe-thalamus and hippocampus-thalamus (U = 77, p = 0.022; U = 87, p = 0.040; U = 81, p = 0.026;

**Table 3. Thyroid test results (n = 13).**

| Test item | Result (mean ± sd) | Reference |
|---|---|---|
| Total thyroxine (T4) | 2.06 ± 0.37 | 1.0–3.2 μg/dL |
| Thyroid stimulating hormone (TSH) | <0.058 ± 0.03 | <0.55 ng/mL |
| Free triiodothyronine (fT3) | 3.84 ± 1.28 | 2.5–7.8 pmol/L |
| Free thyroxine (fT4) | 1.32 ± 0.62 | 0.6–3.0 ng/dL |
| Thyroglobulin test | Negative | Negative |

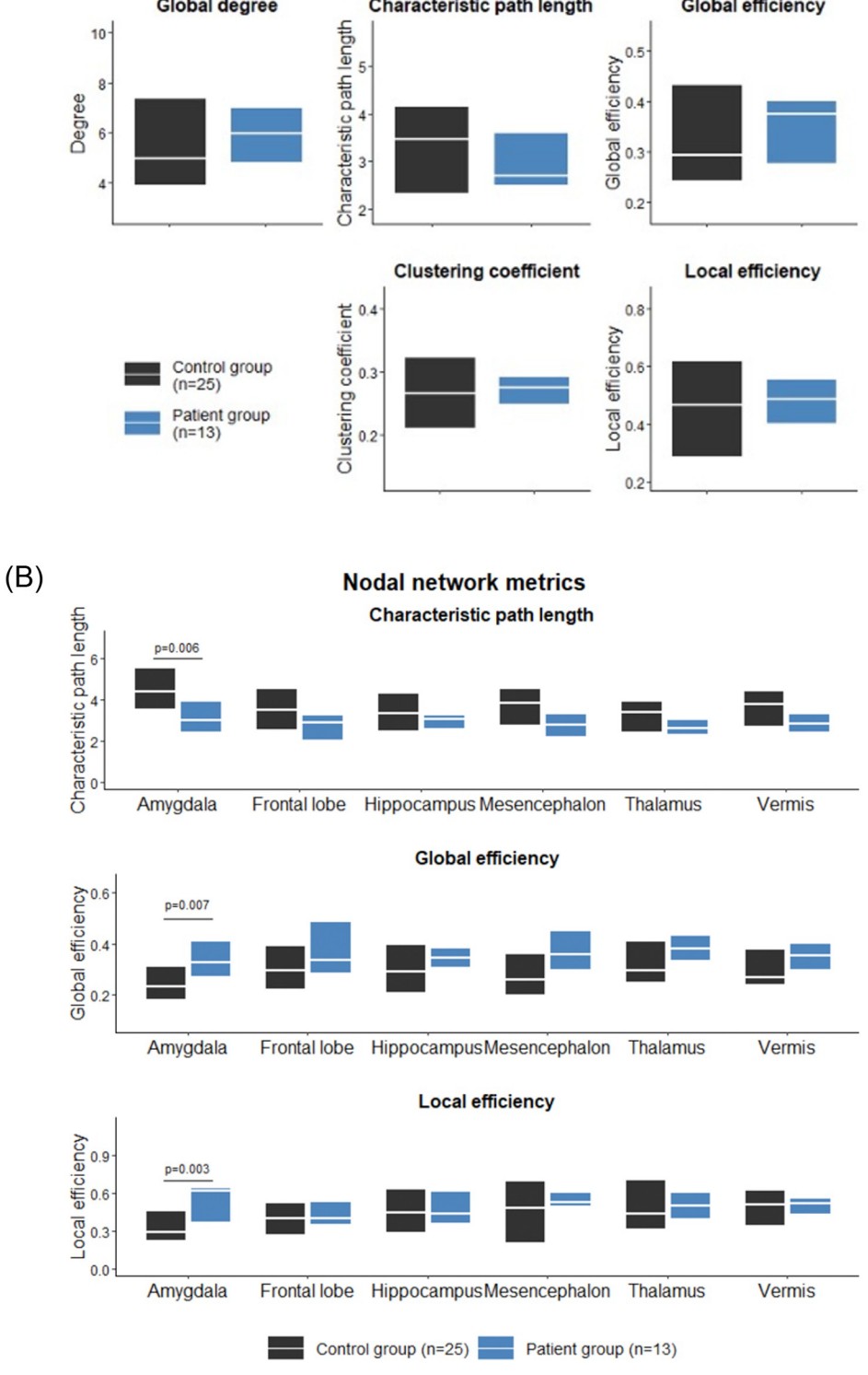

**Fig 4.** A. Global network metrics visualized as a boxplot with median and interquartile range. B. Nodal network metrics.

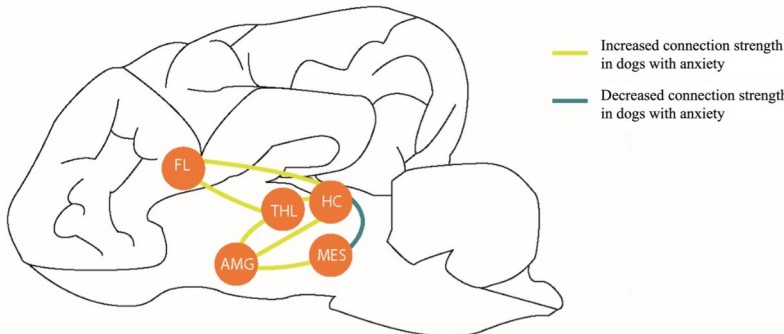

**Fig 5. Connections of the anxiety circuit that differ significantly between the patient group and control group.**
Connections that are significantly higher in the patient group are indicated in yellow, connections that are significantly lower are indicated in blue. Abbreviations: AMG, amygdala; FL, frontal lobe; HC, hippocampus; MES, mesencephalon and THL, thalamus.

U = 64, p = 0.009; U = 78, p = 0.028 and U = 52.5, p = 0.005, respectively). The connection strength between hippocampus and mesencephalon is significantly lower in the patient group compared to the control group (U = 66, p = 0.014). The connections in the anxiety circuit that differed significantly between the patient group and the control group are shown in Fig 5.

### Correlations between network metrics and anxiety symptoms

The demographic information and C-BARQ scores of patients are provided in Table 4. Correlations of global network metrics, nodal degree, global efficiency and clustering coefficient, and connection strength with anxiety symptoms were assessed using Spearman's rank correlation coefficient (Fig 6). Degree ($\rho$ = 0.594, p = 0.049) and global efficiency ($\rho$ = 0.583, p = 0.047) in amygdala were positively correlated with stranger-directed fear. Degree ($\rho$ = -0.593, p = 0.042) in amygdala and connection strength ($\rho$ = -0.781, p = 0.042) between the left and right amygdala were negatively correlated with excitability. Degree ($\rho$ = 0.616, p = 0.033) in hippocampus and connection strength between amygdala and hippocampus ($\rho$ = 0.758, p = 0.028) and amygdala and thalamus ($\rho$ = 0.905, p<0.001) were positively correlated with attachment/attention-seeking behavior. Degree in the hippocampus was positively correlated with trainability ($\rho$ = 0.579, p = 0.049) and touch sensitivity ($\rho$ = 0.837, p = 0.001). Clustering coefficient in the frontal lobe was positively correlated with chasing ($\rho$ = 0.784, p = 0.003). Finally, characteristic path length ($\rho$ = -0.879, p = 0.009) was negatively correlated with familiar dog aggression and local efficiency ($\rho$ = 0.805, p = 0.029), global efficiency ($\rho$ = 0.879, p = 0.009), degree ($\rho$ = 0.954, p = 0.001) in the frontal lobe and global efficiency ($\rho$ = 0.954, p = 0.001) in frontal lobe were positively correlated with familiar dog aggression. For Trainability, the higher score means better trainability. For the rest, the higher score means worse behavior problems.

## Discussion

### Global network topology

At the global network topology level, no significant differences between groups were observed. In humans, many studies reported significant differences between healthy controls and anxiety patients: an increased AUC (area under the curve) of shortest path length and a decreased AUC of clustering coefficient were found in the patients with SAD on a global level [44];

**Table 4. The C-BARQ scores of patients.**

| No | Age (m) | Breed | Gender | Trainability | Stranger-directed aggression | Owner-directed aggression | Dog-directed aggression | Familiar dog aggression | Chasing | Stranger-directed fear | Nonsocial fear | Separation-related problems | Touch sensitivity | Excitability | Attachment/attention-seeking | Energy |
|---|---|---|---|---|---|---|---|---|---|---|---|---|---|---|---|---|
| 1 | 64 | Jack Russell terrier | FC | 1.50 | 0.80 | 0.00 | 2.63 | 0.50 | 3.50 | 1.50 | 1.83 | 0.00 | 0.50 | 2.00 | 1.67 | 2.00 |
| 2 | 66 | Belgian shepherd | MC | 2.25 | 0.00 | 0.63 | 2.88 | 1.00 | 3.50 | 0.00 | 0.17 | 0.00 | 0.75 | 2.83 | 0.17 | 2.00 |
| 3 | 71 | Galgo Espanol | FC | 1.75 | 0.00 | 0.38 | 0.00 | 2.50 | 1.75 | 4.00 | 3.83 | 0.13 | 0.25 | 1.17 | 3.00 | 1.00 |
| 4 | 64 | White Swiss Shepherd | FC | 3.50 | 3.20 | 0.00 | 2.50 | 0.00 | 2.50 | 0.75 | 0.67 | 0.88 | 0.50 | 1.67 | 3.17 | 2.00 |
| 5 | 111 | Akita Inu | MC | 2.88 | 1.00 | 0.25 | 1.25 | ND | 2.25 | 1.50 | 0.00 | 0.00 | 1.50 | 1.00 | 2.00 | 0.00 |
| 6 | 90 | Labrador retriever | MC | 1.50 | 0.00 | 0.00 | 0.75 | ND | 1.00 | 3.00 | 2.83 | 0.00 | 0.50 | 2.00 | 0.33 | 0.00 |
| 7 | 41 | Spanish water dog | MC | 3.63 | 2.30 | 0.00 | 1.00 | ND | 1.50 | 0.00 | 0.50 | 0.50 | 1.00 | 2.33 | 2.67 | 0.00 |
| 8 | 86 | Galgo Espanol | MC | 0.13 | 0.00 | 0.00 | 4.00 | 0.00 | 1.25 | 3.75 | 3.83 | 0.00 | 0.50 | 1.67 | 0.17 | 1.00 |
| 9 | 55 | Belgian shepherd | FC | 3.75 | 0.70 | 0.00 | 0.13 | ND | 1.00 | 1.50 | 2.67 | 0.00 | 1.00 | 2.33 | 3.67 | 4.00 |
| 10 | 98 | Border collie | MC | 2.13 | 0.00 | 0.00 | 1.00 | ND | 2.00 | 0.50 | 2.00 | 0.13 | 1.00 | 2.67 | 1.67 | 1.50 |
| 11 | 57 | American hairless terrier | FC | 3.75 | 0.00 | 0.00 | 0.75 | 0.50 | 2.25 | 1.75 | 2.67 | 0.00 | 1.25 | 2.00 | 3.67 | 2.00 |
| 12 | 151 | Jack Russell terrier | MC | 1.38 | 0.30 | 0.25 | 2.50 | 0.00 | 0.25 | 1.25 | 0.67 | 1.88 | 3.25 | 2.67 | 3.00 | 2.00 |
| 13 | 39 | English cocker spaniel | FC | 2.63 | 2.00 | 0.00 | 2.50 | 0.50 | 2.75 | 3.00 | 1.00 | 0.25 | 1.75 | 2.33 | 2.17 | 2.00 |

Note: FC—castrated female; MC—castrated male; ND–not detectable

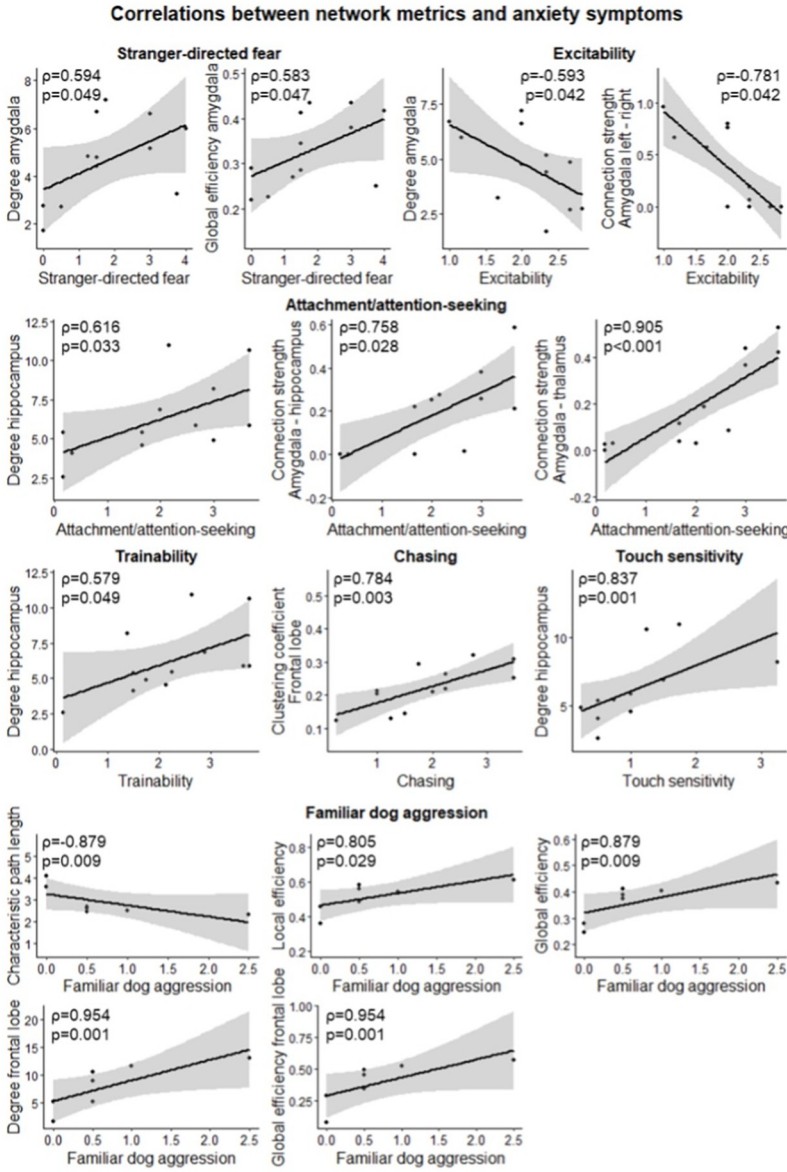

**Fig 6. Correlations of global network metrics, nodal degree, global efficiency and clustering coefficient, and connection strength with anxiety symptoms.** Data are visualized as a scatter plot with regression line and 95% confidence interval.

significantly increased network segregation was observed in GAD [45], and sub-optimal brain-wide organization and integration was present in patients with GAD [46].

Interestingly, a number of studies have also failed to find any global network differences in anxiety disorders: the network structure and node centrality metrics did not differ between the SAD and healthy groups [47]; the global network strength of anxiety symptoms did not change significantly in eating disorder psychopathology before and after treatment [48]. Similarly, in our results global network topology is not significantly different in dogs with anxiety compared to healthy dogs. Nonetheless, in the regional topology level, enhanced functional integration was found, which may indicate greater resilience to focal neural damage in the brain.

Brain network topological resilience has been assumed to protect the integrity of the network from pathological attack [49]. It might explain why there was no significant difference in the global level of our study. Only a few specific regions are affected, and this may not influence the global network. But it can also be directly related to anxious behavior traits. For instance, there is a considerable body of evidence that amygdala output is directly related to fear and anxiety phenotypes [50, 51].

## Regional topological disorganization of functional networks in anxious dogs

In the anxiety group, characteristic path length was found to be lower and global efficiency higher in the amygdala. This might indicate that there is better communication efficiency between these regions and the rest of the network when confronted with anxiety. More strongly connected and better communication efficiency can be regarded as enhanced functional integration for information transfer between these brain regions and the rest of the network. In this study, we also detected that the local efficiency is higher in the amygdala in anxiety dogs when compared with the healthy group. This indicates that there is more clustering around the amygdala, so a better local interconnectivity in the anxious dogs. Increased local efficiency also indicates increased functional segregation in the amygdala. For connection strength between connections of the anxiety circuit, a more (efficient) communication in the patient group between amygdala-mesencephalon and amygdala-thalamus were observed. Here, the increased FC of these three regions is in line with a well-accepted hypothesis that anxiety and anxiety disorders are associated with increased or overactive functioning in the salience network [3].

Furthermore, in humans, it is reported that the default mode network (DMN) may interact with other brain networks during emotion regulation. For example, individuals with high trait anxiety demonstrate decreased FC between regions of the DMN and the frontoparietal network [52]. In this study, an increased connection strength between the frontal lobe and hippocampus was found. Interestingly, Hang X et al. reported that aberrant FC of some crucial brain regions of the DMN and the salience network might contribute to the pathophysiology of anxiety disorders in humans [53]. Correspondingly, in our current study, FC between amygdala-hippocampus, mesencephalon-hippocampus and thalamus-hippocampus was increased. Another hypothesis is that conflicting signals generated in the salience network are relayed to the frontoparietal network that implements increased cognitive control on future trials [3]. This corresponds with the increased connectivity between the frontal lobe and the thalamus in our findings.

A particular highlight of our results is the connection between the hippocampus and mesencephalon. Here, a less (efficient) communication was found between hippocampus and mesencephalon in the anxiety group. Of note, current findings indicate that the hippocampus and the mesencephalon are seen as partners in "integrative encoding", suggesting that the neurotransmitter dopamine may be involved. This points toward an exciting synthesis among cognitive-, molecular-, and systems-level memory research with implications for clinical conditions in which dopaminergic neuromodulation is dysfunctional [54]. It has been reported that dysfunction of the hippocampus and the mesencephalon is related with high risk for psychosis in humans [55], and in rats destruction of dopaminergic neurons in the mesencephalon decreases hippocampal cell proliferation, and can be reversed by fluoxetine, suggesting fluoxetine might be potential therapeutic drugs for non-motor symptoms (e.g., anxiety, depression and cognitive deficits) in Parkinson disease [56]. Also in dogs, dopaminergic systems play an important role in determining affective reactions such as the exhibition of anxiety related

behaviour problems [29, 57]. Lisman et al. developed the concept that hippocampus and mid-brain dopaminergic neurons form a functional loop and proved that the enhanced connectivity between hippocampus and mesencephalon is associated with learning and information processing [58]. Thus, in our study, the lower connectivity between hippocampus and mesencephalon in the patient group may be the reason for the decreased trainability symptoms.

### Correlations between network metrics and anxiety symptoms

Several tools have been developed to measure canine behavior. Some are based on the direct observation of the dog's response to several test situations [59]. Although such tests are more objective than owner-derived information, the disadvantage is that it is difficult to evoke problem behavior in a clinical setting [60]. Other methods focus on the assessment of day-to-day behavior using a questionnaire especially designed for the dog owner. The C-BARQ is such a validated and widely used questionnaire for canine behavior.

In this study, stranger-directed fear, excitability, attachment, attention-seeking, trainability, chasing, touch sensitivity and familiar dog aggression were found to be associated with several network measures: increased FC of amygdala corresponded with more stranger-directed fear and lower excitability; increased connectivity of amygdala, hippocampus and thalamus associated with more attachment and attention-seeking behavior; increased connectivity of hippocampus correlated with a better trainability and a higher touch sensitivity; increased connectivity of frontal lobe corresponds with more chasing; increased global and local efficiency, correlated with worse familiar dog aggression; increased functional connectivity of frontal lobe also corresponded with worse familiar dog aggression.

In the dog behavior, the amygdala and hippocampus are associated with remembering things and getting aroused, excited and scared [61]. Dysfunctions of these regions can lead to anxiety symptoms like more fear, less excitability, less trainability and so on, which are in line with previous human research [62, 63]. Vermeire et al. already reported that aberrant thalamus function was observed in compulsive dogs [29]. In this study, we also observed abnormal behavior of attachment/attention-seeking is associated with the thalamus, which is consistent with previous findings of anxiety symptoms in humans [64, 65]. The findings of more chasing and worse familiar dog aggression related to the frontal lobe are also in line with human studies that found that frontal dysfunction predicts depression and anxiety symptoms [66, 67].

### Limitations

Several limitations in the present study need to be considered. First, the sample size is relatively moderate. 13 dogs with different backgrounds and symptom severity might be insufficient to find global network differences. Disagreeable experience should be considered in studies examining the relationship between network differences and psychopathology [68, 69]. In this study, most patient dogs are adopted from the animal shelter, maltreatment or jettison may influence the anxiety brain networks; and a few were raised by the owners. More patients should be recruited to draw a clearer result. Second, the difference between lab and domestic dogs should be mentioned [70]. With the current data it will be impossible to disentangle the effect of 'housing' on the anxiety symptoms used in this cohort. We can only add to the future directions that it would be advised to research canine anxiety symptoms as well in lab animals and/or to use a more naturalistic healthy canine control group. This can be refined in the future when we have enough patients then we may draw a clear difference or consistency within the group; then beagle patients can make a more direct comparison with the control group. Third, our MRI protocol was performed under anesthesia. This might cause deviations with the awake condition. Fourth, that we have included the entire prefrontal as ROI should

be considered as a limitation, given that this is a large region including subregions with potentially different functionalities in the dog anxiety system, although this is at this moment not yet established in the dog.

## Conclusion

By using rs-fMRI and graph theory network analysis in dogs, we characterized abnormalities in resting-state functional brain network topology associated with anxiety. As we found correlations between anxiety symptoms and network measures, this may indicate that rs-fMRI could provide useful diagnostic information for anxiety in dogs, although further research is still required. In the future, we would also like to investigate the potential of rs-fMRI as a diagnosis tool for treatment response, such as pharmacological treatments or neural modulation treatments like rTMS. Such efforts will provide important insight into pathophysiological mechanisms of anxiety in dogs, which can lead to more personalized and effective therapies, and together with other animal research, build a bridge to the understanding of human behavior (and vice versa). Further work in larger sample sizes is needed to substantiate our observations that specific brain connectivities are associated with anxiety in dogs.

## Supporting information

**S1 Text. Monitoring of welfare in dogs kept and used for research purposes by the Ghent University Ethical Committee".**
(PDF)

**S1 Fig. The global efficiency (a), local efficiency (b), and characteristic path length (c) of amygdala in a range of sparsity thresholds (20%-50%, with 5% intervals)".**
(DOCX)

## Acknowledgments

We would like to thank the department of medical imaging and small animal orthopedics, Ghent University to provide access to their database and facilities. The authors would like to thank Ir. Gert Vanhollebeke for his excellent contribution during the revision work.

## Author Contributions

**Conceptualization:** Kathelijne Peremans, Jimmy H. Saunders, Christian Vanhove, Chris Baeken.

**Data curation:** Yangfeng Xu, Emma Christiaen, Sara De Witte, Qinyuan Chen.

**Formal analysis:** Yangfeng Xu, Emma Christiaen.

**Investigation:** Yangfeng Xu, Emma Christiaen, Sara De Witte, Qinyuan Chen.

**Methodology:** Yangfeng Xu, Emma Christiaen, Sara De Witte, Qinyuan Chen.

**Project administration:** Kathelijne Peremans, Chris Baeken.

**Resources:** Kathelijne Peremans, Chris Baeken.

**Software:** Yangfeng Xu, Emma Christiaen, Jimmy H. Saunders, Christian Vanhove.

**Supervision:** Kathelijne Peremans, Jimmy H. Saunders, Christian Vanhove, Chris Baeken.

**Validation:** Sara De Witte, Qinyuan Chen.

**Visualization:** Yangfeng Xu, Emma Christiaen.

**Writing – original draft:** Yangfeng Xu, Emma Christiaen.

**Writing – review & editing:** Sara De Witte, Qinyuan Chen, Kathelijne Peremans, Jimmy H. Saunders, Christian Vanhove, Chris Baeken.

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
