## [Decision Letter · Decision Letter 0]

19 Apr 2022

PONE-D-22-02320Network analysis reveals abnormal functional brain circuitry in anxious dogsPLOS ONE

Dear Dr. Xu,

Thank you for submitting your manuscript to PLOS ONE. After careful consideration, we feel that it has merit but does not fully meet PLOS ONE’s publication criteria as it currently stands. Therefore, we invite you to submit a revised version of the manuscript that addresses the points raised during the review process.

We look forward to receiving your revised manuscript.

Kind regards,

Tamas Kozicz

Academic Editor

PLOS ONE

Journal Requirements:

“This study is funded by Belgium governmental FWO institution (Project number G011018N). Emma Christiaen is an SB PhD fellow at Research Foundation - Flanders (Project number 1S90218N).”

“NO. The funders had no role in study design, data collection and analysis, decision to publish, or preparation of the manuscript.”

Additional Editor Comments (if provided):

Dear Dr. Yangfeng Xu

Thank you for submitting your manuscript to PLOS One.

I have completed the evaluation of your manuscript. The reviewers recommend reconsideration of your manuscript following substantial revision. I invite you to resubmit your manuscript after addressing the reviewers’ comments.

When revising your manuscript, please consider all issues mentioned in the reviewers' comments carefully: please outline every change made in response to their comments and provide suitable rebuttals for any comments not addressed. Please note that your revised submission may need to be re-reviewed. 

PLOS One values your contribution and I look forward to receiving your revised manuscript.

Sincerely Yours,

Tamas Kozicz

Reviewers' comments:

Reviewer's Responses to Questions

**Comments to the Author**

1. Is the manuscript technically sound, and do the data support the conclusions?

Reviewer #1: Partly

Reviewer #2: Partly

2. Has the statistical analysis been performed appropriately and rigorously? 

Reviewer #1: No

Reviewer #2: No

3. Have the authors made all data underlying the findings in their manuscript fully available?

Reviewer #1: Yes

Reviewer #2: Yes

4. Is the manuscript presented in an intelligible fashion and written in standard English?

Reviewer #1: Yes

Reviewer #2: Yes

5. Review Comments to the Author

Reviewer #1: Xu et al. set out to investigate potential abnormalities in function brain networks in anxious dogs. They assessed network topology using resting-state fMRI scanning and graph theory to compare healthy dogs vs. patients. Overall, the study is interesting and well-written, but should be checked on occasional grammatical errors. Furthermore, I have some general questions and suggestions for further improvement.

Main:

1) The authors picked 5 brain regions to study the ‘anxiety circuit’. It would be good to include a rationale for including these regions.

Also, ‘frontal lobe’ is rather unspecific, as both rodent and human work indicates subregional specificity within the frontal cortex in the modulation of anxiety, with some subregions boosting and others suppressing anxiety. Why did the authors decide to take this rather unspecific readout?

2) Furthermore, the data don’t really convince me about the specificity of the reported effects to the anxiety circuit. Overall, effects seen in nodal network metrics are also found on the global level, but there they only fail to reach significance. As the authors run tests on all of the brain regions independent of each other, the effects on these regions are not directly compared, making that specificity cannot be claimed. Would there be a ‘control’ region that could be taken along to compare the effects against? Or could the authors maybe include all regions as within subject factor, to test whether all regions were similarly affected or whether effects were region-specific? The effects in the amygdala at least seem to be robust, and differ in same cases (clustering coefficient) from the other regions?

3) Related to point 2, is the number of comparisons made by the authors and the corrections for multiple testing that are actually performed. Whereas the authors do report on FDR correction for connection readouts and connectivity correlations, they do not correct any of the other readouts for the comparisons made. For example Fig 3B only already contains 25 comparisons, and none of them seems corrected for multiple testing. Similarly, Fig 5 reports on significant correlations that are not corrected for multiple testing, except for the connectivity readout.

Minor:

4) It is not completely clear to me what the exact rationale is for the study. Is it to better understand and treat anxiety in dogs, or to develop an animal model for anxious patients to allow for more invasive recordings/manipulations? The latter is mainly done in rodents, and if the authors think their model is superior to this, it would be good to further explain their reasoning. Currently, the MS includes both an introduction on human anxiety and that in dogs; the work might benefit from making the goal very clear and tailor the introduction and discussion towards this goal.

5) The manuscript would benefit from including a brief explanation of the distinct readouts of graph theory in a layman’s style in the introduction or results section. Terminology such as global efficiency, path length and nodal degree is difficult to grasp for non-experts, whereas the findings might also be of interest to them.

Reviewer #2: In the present manuscript, Xu et al. investigated functional brain network topology changes in anxious dogs (n=13) compared to healthy dogs (n=25). While their findings indicate no modifications at the whole brain level, the analysis focused on the anxiety circuit highlighted network topological changes at the node level, which resulted to be correlated with anxiety symptoms.

Despite I found the manuscript interesting, I have some concerns that the Authors should address.

- Did the Authors perform field inhomogeneity-related artifact correction?

- Were the global WM and CSF signal regressed in fMRI processing?

- Did the Authors check whether the brain connectomes exhibit a small-world behavior in line with current knowledge on brain networks? This should be verified considering the relative low number of nodes in the networks

- How did they deal with negative weights in brain connectomes?

- Have they checked if there are disconnected nodes in the functional connectomes after the thresholding procedure?

- Did the authors compute the network measures on the weighted or binarized networks?

- P.11, line 204, degree and strength are two different measures. Did the authors compute the degree or strength? Or Both? Throughout the manuscript they refer to the degree.

- More details, formula and related references should be reported for the computation of the network measures

- Did they compute the network measures for all the nodes or only those involved in the anxiety circuit? This should be better clarified in the methods section.

- Did the authors take into account multiple comparisons correction for the node-level analysis? Are the reported -pvalues uncorrected? I feel that some form of correction is warranted to ensure that nodes exhibiting different nodal topology do not suffer from multiple statistical tests (across different nodes and across different nodal measures).

- Why did they average left and right in node-level analysis of the anxiety circuit? It would be interesting to assess whether hemispheric differences exist

- In the conclusion section, the Authors state that “rs-fMRI could be used as a biomarker for anxiety”, I would suggest toning down this section as no conclusive evidence on this direction can be drawn from the present study. Correlation findings, especially on such a small sample, are not indicative of potential biomarkers.

6. PLOS authors have the option to publish the peer review history of their article (what does this mean?). If published, this will include your full peer review and any attached files.

Reviewer #1: No

Reviewer #2: No

---

## [Author Response · Author response to Decision Letter 0]

10 Aug 2022

Comments from Reviewer 1

Main:

1) The authors picked 5 brain regions to study the ‘anxiety circuit’. It would be good to include a rationale for including these regions.

Also, ‘frontal lobe’ is rather unspecific, as both rodent and human work indicates subregional specificity within the frontal cortex in the modulation of anxiety, with some subregions boosting and others suppressing anxiety. Why did the authors decide to take this rather unspecific readout?

We thank the referee for these insightful comments.

First, we picked these five regions based on our former research in behavior-disordered dogs, and the known human counterparts, since no clear information is available regarding the anxiety circuitry in dogs. To clarify, we have added following information in the method section:

Line 234-237: Based on our former research [8, 13, 26, 27, 38-40], relevant veterinary behavior medicine research [41], and human medicine [28-31], we choose these five regions of interest as the anxiety circuit in this study: amygdala, frontal lobe, hippocampus, mesencephalon, and thalamus.

For the ‘frontal lobe’, however, we must mention that, in contrast to rodents, in the anxiety canine model clearly showing behavior abnormalities little is known about specific anxiety subcircuits in the frontal cortical areas. Therefore, we had decided to not subdivide the frontal cortex, given that the frontal cortex by itself is not the focus of this research. Nevertheless, we acknowledge this as a study limitation, so we have added the following sentence to the discussion section:

Line 458-461: Fourth, that we have included the entire prefrontal as ROI should be considered as a limitation, given that this is a large region including subregions with potentially different functionalities in the dog anxiety system, although this is at this moment not yet established in the dog.

2) Furthermore, the data don’t really convince me about the specificity of the reported effects to the anxiety circuit. Overall, effects seen in nodal network metrics are also found on the global level, but there they only fail to reach significance. As the authors run tests on all of the brain regions independent of each other, the effects on these regions are not directly compared, making that specificity cannot be claimed. Would there be a ‘control’ region that could be taken along to compare the effects against? Or could the authors maybe include all regions as within subject factor, to test whether all regions were similarly affected or whether effects were region-specific? The effects in the amygdala at least seem to be robust, and differ in same cases (clustering coefficient) from the other regions?

Fair points. 

As requested, we added vermis as ‘control’ region in the nodal level analysis. Therefore, between-group differences in the nodal level were assessed in the regions of anxiety circuit (including the frontal lobe, hippocampus, thalamus, amygdala, and mesencephalon), and we added the vermis as a ‘control’ region using non-parametric Mann-Whitney U test. To correct the network parameters for multiple comparisons, we applied FDR correction by using p < .05 as the significant threshold. We found that the patient group exhibited significantly increased global efficiency and local efficiency in the amygdala (p = 0.007, FDR correction; p = 0.003, FDR correction, respectively), compared with the control group. We also found that the patient group exhibited significantly decreased characteristic path length in the amygdala (p = 0.006, FDR correction), compared with the control group.

Consequently, we have revised the manuscript accordingly with these more stringent corrections.

In method section:

Line 237-243: The vermis was chosen as the control region. Therefore, between-group differences in the nodal level were assessed in the regions of the anxiety circuit and vermis. The nonparametric Mann-Whitney U test was performed in the nodal level analysis to compare the between-group differences. To correct the network parameters for multiple comparisons, false discovery rate (FDR) correction was applied by using p < .05 as the significant threshold. 

In results section:

Line 288-292: In the amygdala, we found that the patient group exhibited significantly increased global efficiency and local efficiency (p = 0.007, FDR correction; p = 0.003, FDR correction, respectively), and significantly decreased characteristic path length (p = 0.006, FDR correction), compared with the control group.

3) Related to point 2, is the number of comparisons made by the authors and the corrections for multiple testing that are actually performed. Whereas the authors do report on FDR correction for connection readouts and connectivity correlations, they do not correct any of the other readouts for the comparisons made. For example Fig 3B only already contains 25 comparisons, and none of them seems corrected for multiple testing. Similarly, Fig 5 reports on significant correlations that are not corrected for multiple testing, except for the connectivity readout.

Thank you for this remark. Normally in graph analysis, we only corrected for multiple comparisons between regions, not between parameters. As we answered in point 2, at nodal level, we found that only in the amygdala, the global efficiency, local efficiency and characteristic path length were significantly different between patient and control groups. If we corrected for 25 comparisons, there were no significantly difference.

Minor:

4) It is not completely clear to me what the exact rationale is for the study. Is it to better understand and treat anxiety in dogs, or to develop an animal model for anxious patients to allow for more invasive recordings/manipulations? The latter is mainly done in rodents, and if the authors think their model is superior to this, it would be good to further explain their reasoning. Currently, the MS includes both an introduction on human anxiety and that in dogs; the work might benefit from making the goal very clear and tailor the introduction and discussion towards this goal.

The goal of this study is bifold: anxiety in dogs and natural animal model. Rodent model research is okay but does not provide a natural model as in general it is conducted on genetical, pharmacological, physical, manipulated animals.

Consequently, we have adapted some parts in the introduction and discussion.

In introduction section:

Line 69-71: Thus, dogs might be a better model to investigate the mechanisms of anxiety as the rodent research is in general conducted on genetical, pharmacological, physical, manipulated animals.

Line 115-117: The results in dogs will benefit both veterinary medicine for anxiety-disordered animals and may serve human medicine as a natural model.

In conclusion section:

Line 469-472: Such efforts will provide important insight in pathophysiological mechanisms and anxiety illness course in dogs that may lead to more personalized and effective therapies and provide a natural animal model for human medicine.

5) The manuscript would benefit from including a brief explanation of the distinct readouts of graph theory in a layman’s style in the introduction or results section. Terminology such as global efficiency, path length and nodal degree is difficult to grasp for non-experts, whereas the findings might also be of interest to them.

Agreed. We have added some information in the introduction section about graph theory parameters, and Table 3.

Line 103-109: Specifically, nodal degree measures the degree of nodes tending to cluster together, global efficiency measures the efficiency of parallel information transfer through the network, clustering coefficient measures the efficiency of information exchange within a local subnetwork or among adjacent regions, characteristic path length measures the ability for information propagation within the network, the small-worldness indicates a typical network that has similar path length but higher clustering than a random network [18-20].

Line 243-244: The calculation of network measures of integration, segregation and centrality is summarized in Table 3 [18].

Comments from Reviewer 2

1 Did the Authors perform field inhomogeneity-related artifact correction?

Thank you for this point. We didn’t perform field inhomogeneity-related artifact correction since this is not typically done during preprocessing of rsfMRI. However, we did perform 2nd order shimming before image acquisition to optimize the field homogeneity. This point has been added to the revision.

2 Were the global WM and CSF signal regressed in fMRI processing?

Good point. We did not do global WM and CSF signal regression. Regression of these signals is considered controversial, with some researchers believing that it leads to loss of information [26]. Because of this controversy, we decided not to regress out global WM and CSF signals. This point has been added to the revision.

[26] Grajauskas, L. A., Frizzell, T., Song, X., & D’Arcy, R. C. (2019). White matter fMRI activation cannot be treated as a nuisance regressor: Overcoming a historical blind spot. Frontiers in neuroscience, 13, 1024.

3 Did the Authors check whether the brain connectomes exhibit a small-world behavior in line with current knowledge on brain networks? This should be verified considering the relative low number of nodes in the networks

Thank you for this remark. The small-world metrics normalized clustering coefficient, normalized characteristic path length and small-worldness are similar to those of the macaque and mouse brain, that exhibit small-world behavior [19]. We found σ > 1 for the two groups, indicating that both patient and control groups exhibited small-world. Thus, we have added the point in the revision:

Line 230-232: Small-worldness (σ) indicates a typical network that has similar path length but higher clustering than a random network [19].

Line 279-281: In addition, we found σ > 1 for the two groups, indicating that both patient and control groups exhibited small-world attributes.

[19] Bassett, D. S., & Bullmore, E. T. (2017). Small-world brain networks revisited. The Neuroscientist, 23(5), 499-516.

4 How did they deal with negative weights in brain connectomes?

Only positive weights were included, negative weights were disregarded, as is typically done in graph theoretical analyses. 

5 Have they checked if there are disconnected nodes in the functional connectomes after the thresholding procedure?

It was checked. All nodes have a degree higher than 1, so no disconnected nodes, although we still might have disconnected clusters.

6 Did the authors compute the network measures on the weighted or binarized networks?

Thanks for this point. We computed the network measures on the weighted networks. To make it clear, we have added this point in the revision.

7 P.11, line 204, degree and strength are two different measures. Did the authors compute the degree or strength? Or both? Throughout the manuscript they refer to the degree.

We computed the strength. We adapted this sentence in the revision as:

Connection strength is the number of (weighted) edges connected to a node.

8 More details, formula and related references should be reported for the computation of the network measures

We have added a new table describing the computation of the network measures in the method section.

9 Did they compute the network measures for all the nodes or only those involved in the anxiety circuit? This should be better clarified in the methods section.

We computed the network measures for all the nodes. As requested, we added this into the revision.

For remarks 1,2, 4, 6, and 9, we have added these points in the method part:

Line 194-197:

During the processing, field inhomogeneity-related artifact correction was not performed, global white matter (WM) and cerebrospinal fluid (CSF) were not regressed [26], only positive weights in brain connectomes were included, and the network measures for all the nodes were computed on the weighted networks.

10 Did the authors take into account multiple comparisons correction for the node-level analysis? Are the reported -pvalues uncorrected? I feel that some form of correction is warranted to ensure that nodes exhibiting different nodal topology do not suffer from multiple statistical tests (across different nodes and across different nodal measures).

Yes, we had corrected for multiple comparisons between nodes in the anxiety circuit. We have clarified this better in the revision.

Line 241-243:

To correct the network parameters for multiple comparisons, FDR correction was applied by using p < .05 as the significant threshold.

11 Why did they average left and right in node-level analysis of the anxiety circuit? It would be interesting to assess whether hemispheric differences exist

We didn’t assess whether hemispheric differences exist in this study. According to a rat study of our team with similar processing procedures, changes were very similar left and right (Christiaen et al., 2019). The advantage is that if there are less comparisons, less stringent correction necessary.

CHRISTIAEN, E., GOOSSENS, M.-G., RAEDT, R., DESCAMPS, B., LARSEN, L. E., CRAEY, E., CARRETTE, E., VONCK, K., BOON, P. & VANHOVE, C. 2019. Alterations in the functional brain network in a rat model of epileptogenesis: A longitudinal resting state fMRI study. Neuroimage, 202, 116144.

12 In the conclusion section, the Authors state that “rs-fMRI could be used as a biomarker for anxiety”, I would suggest toning down this section as no conclusive evidence on this direction can be drawn from the present study. Correlation findings, especially on such a small sample, are not indicative of potential biomarkers.

Indeed, the former conclusion we made was a bit ambitious. We have adapted the conclusion section in this revision:

Line 465-469: As we found correlations between anxiety symptoms and network measures, this may indicate that rs-fMRI could provide useful diagnostic information for anxiety in dogs, although further research is still required. In the future, we would also like to investigate the potential of rs-fMRI as a diagnosis tool for treatment response, such as pharmacological treatments or neural modulation treatments like rTMS.

---

## [Decision Letter · Decision Letter 1]

12 Sep 2022

PONE-D-22-02320R1Network analysis reveals abnormal functional brain circuitry in anxious dogsPLOS ONE

Dear Dr. Xu,

Thank you for submitting your manuscript to PLOS ONE. After careful consideration, we feel that it has merit but does not fully meet PLOS ONE’s publication criteria as it currently stands. Therefore, we invite you to submit a revised version of the manuscript that addresses the points raised during the review process.

We look forward to receiving your revised manuscript.

Kind regards,

Tamas Kozicz

Academic Editor

PLOS ONE

Reviewers' comments:

Reviewer's Responses to Questions

**Comments to the Author**

1. If the authors have adequately addressed your comments raised in a previous round of review and you feel that this manuscript is now acceptable for publication, you may indicate that here to bypass the “Comments to the Author” section, enter your conflict of interest statement in the “Confidential to Editor” section, and submit your "Accept" recommendation.

Reviewer #1: (No Response)

Reviewer #2: (No Response)

2. Is the manuscript technically sound, and do the data support the conclusions?

Reviewer #1: Yes

Reviewer #2: Partly

3. Has the statistical analysis been performed appropriately and rigorously? 

Reviewer #1: Yes

Reviewer #2: Yes

4. Have the authors made all data underlying the findings in their manuscript fully available?

Reviewer #1: No

Reviewer #2: No

5. Is the manuscript presented in an intelligible fashion and written in standard English?

Reviewer #1: Yes

Reviewer #2: No

6. Review Comments to the Author

Reviewer #1: The authors have answered most of my comments.

However, one of my previous points is not yet optimally dealt with:

"4) It is not completely clear to me what the exact rationale is for the study. Is it to better

understand and treat anxiety in dogs, or to develop an animal model for anxious

patients to allow for more invasive recordings/manipulations? The latter is mainly done

in rodents, and if the authors think their model is superior to this, it would be good to

further explain their reasoning. Currently, the MS includes both an introduction on

human anxiety and that in dogs; the work might benefit from making the goal very clear

and tailor the introduction and discussion towards this goal."

The authors state in their answer that their goal is bifold; anxiety in dogs and natural animal model. I totally agree on anxiety in dogs, but the statements on rodent research are rather blunt. Rodents may be evolutionary more distinct from humans than dogs, but the manipulations listed are not always implemented, and if applied, serve specific purposes such as targeted manipulations of cells or circuits to study their effects on behavior. Studies in dogs have other flaws; for example that there is much less knowledge to build on than in rodents, and there might be more stringent ethical restrictions to work in dogs (and less possibilities for manipulations to study causality). According to the 3R principle the work in dogs should have clear benefits over working with other animal models, to warrant their use. As such, I am not convinced by the current answer and text added to the MS. A more nuanced discusion would benefit the paper.

Reviewer #2: I thank the Authors for addressing most of my concerns. I have a few comments yet.

1. Was the C-BARQ administered both to healthy and anxious dogs?

2. Lines 196-197, I do not understand way the Authors chose to report “only positive weights in brain connectomes were included, and the network measures for all the nodes were computed on the weighted networks.” in the Preprocessing section. This explanation would better fit the section Functional network construction.

3. Did the Authors check whether the network measures were consistent across the range of thresholds explored?

4. Line 258, should “connections” read “comparisons”?

5. Was the statistical analysis at the connection level performed on the 30x30 connectivity matrix? Or did the authors extract only the connectivity patterns between the five regions of interest? Reading lines 295-296, I assume the latter. In this case, why did the Authors choose to create 30x30 connectivity matrices instead of 5x5 matrices? What is the role of the remaining brain regions? In addition, by doing this, the node-level topological measures reflect not only the of connectivity between the anxiety-related regions, but also the connectivity patterns of such regions with the remaining 25 brain areas. And this may also explain why no changes in global network topology have been identified.

7. PLOS authors have the option to publish the peer review history of their article (what does this mean?). If published, this will include your full peer review and any attached files.

Reviewer #1: No

Reviewer #2: No

---

## [Author Response · Author response to Decision Letter 1]

27 Oct 2022

Reviewer #1: The authors have answered most of my comments.

However, one of my previous points is not yet optimally dealt with:

"4) It is not completely clear to me what the exact rationale is for the study. Is it to better understand and treat anxiety in dogs, or to develop an animal model for anxious patients to allow for more invasive recordings/manipulations? The latter is mainly done in rodents, and if the authors think their model is superior to this, it would be good to further explain their reasoning. Currently, the MS includes both an introduction on human anxiety and that in dogs; the work might benefit from making the goal very clear and tailor the introduction and discussion towards this goal."

The authors state in their answer that their goal is bifold, anxiety in dogs and natural animal model. I totally agree on anxiety in dogs, but the statements on rodent research are rather blunt. Rodents may be evolutionary more distinct from humans than dogs, but the manipulations listed are not always implemented, and if applied, serve specific purposes such as targeted manipulations of cells or circuits to study their effects on behavior. Studies in dogs have other flaws; for example, that there is much less knowledge to build on than in rodents, and there might be more stringent ethical restrictions to work in dogs (and less possibilities for manipulations to study causality). According to the 3R principle the work in dogs should have clear benefits over working with other animal models, to warrant their use. As such, I am not convinced by the current answer and text added to the MS. A more nuanced discussion would benefit the paper.

Thank you for this useful remark. In fact, the realm of our research, is all that. It is to better understand and treat anxiety in dogs and develop an animal model. However, the referee is right that putting it all here in this manuscript is confusing, and it does not help the clarity of the paper. Of course, in this paper the goal is not to prove that dog models are superior to rodent models, as we did not include rodents in our sample. 

We agree with the referee in stating that rodent models can serve better in specific purposes. We only hypothesize, that our canine model can serve as a more natural translational model, together with rodent and other animal models, which can be model for for human anxiety (and vice versa). So, we have toned town our former statement as followed:

Introduction part:

Line 70-72: Thus, the canine species might be an appropriate model to investigate brain networks involved in anxiety, and together with other animal research, such as rodents, can be used as a model for human anxiety (and vice versa).

Discussion part:

Line 470-473: Such efforts will provide important insight into pathophysiological mechanisms of anxiety in dogs, which can lead to more personalized and effective therapies, and together with other animal research, build a bridge to the understanding of human behavior (and vice versa).

Reviewer #2: I thank the Authors for addressing most of my concerns. I have a few comments yet.

1. Was the C-BARQ administered both to healthy and anxious dogs?

No, the C-BARQ was only administered to anxious dogs as mentioned in M&M-Patient recruitment. The healthy beagles got regular check by the caretakers and veterinarians, as mentioned in M&M-Animal.

2. Lines 196-197, I do not understand way the Authors chose to report “only positive weights in brain connectomes were included, and the network measures for all the nodes were computed on the weighted networks.” in the Preprocessing section. This explanation would better fit the section Functional network construction.

Good point. We have moved this to the Functional network construction section in line 208-209.

3. Did the Authors check whether the network measures were consistent across the range of thresholds explored?

No, we didn’t check. But probably the network measures were proportional to network density because of the weighted edges, as mentioned in M&M Functional network reconstruction.

4. Line 258, should “connections” read “comparisons”?

Thank you for this point. Yes, it should be “comparisons”. We have adapted it in the manuscript.

5. Was the statistical analysis at the connection level performed on the 30x30 connectivity matrix? Or did the authors extract only the connectivity patterns between the five regions of interest? Reading lines 295-296, I assume the latter. In this case, why did the Authors choose to create 30x30 connectivity matrices instead of 5x5 matrices? What is the role of the remaining brain regions? In addition, by doing this, the node-level topological measures reflect not only the of connectivity between the anxiety-related regions, but also the connectivity patterns of such regions with the remaining 25 brain areas. And this may also explain why no changes in global network topology have been identified.

Fair point. We wanted to look at the effect of anxiety on the whole brain and investigate whether whole-brain connections of the anxiety regions were affected, not just the connections in the anxiety circuit. For our graph theory application, the selected ROIs do not necessarily respect the functional boundaries of the brain [1, 2]. We did 30x30 because we assumed they are involved in anxiety, but we didn’t find global level difference, so, based on the calculated results from 30x30, we selected these 5 key regions for further analysis. On the other hand, if we would have chosen the connectivity matrices 5x5, the rest regions (25 regions) would not have been involved. This would have excluded the rest regions as potential influential nodes in the network.

[1] Yu Q, Du Y, Chen J, et al. Application of graph theory to assess static and dynamic brain connectivity: Approaches for building brain graphs[J]. Proceedings of the IEEE, 2018, 106(5): 886-906.

[2] Khullar S, Michael A M, Cahill N D, et al. ICA-fNORM: Spatial normalization of fMRI data using intrinsic group-ICA networks[J]. Frontiers in systems neuroscience, 2011, 5: 93.

---

## [Decision Letter · Decision Letter 2]

4 Jan 2023

PONE-D-22-02320R2Network analysis reveals abnormal functional brain circuitry in anxious dogsPLOS ONE

Dear Dr. Xu,

Thank you for submitting your manuscript to PLOS ONE. After careful consideration, we feel that it has merit but does not fully meet PLOS ONE’s publication criteria as it currently stands. Therefore, we invite you to submit a revised version of the manuscript that addresses the points raised during the review process.

We look forward to receiving your revised manuscript.

Kind regards,

Tamas Kozicz

Academic Editor

PLOS ONE

Journal Requirements:

Additional Editor Comments (if provided):

While all reviewers agreed that the manuscript improved, one minor issue to be addressed has remained open. Specifically, please clarify why you did not check for network measure consistency across the thresholds.

Reviewers' comments:

Reviewer's Responses to Questions

**Comments to the Author**

1. If the authors have adequately addressed your comments raised in a previous round of review and you feel that this manuscript is now acceptable for publication, you may indicate that here to bypass the “Comments to the Author” section, enter your conflict of interest statement in the “Confidential to Editor” section, and submit your "Accept" recommendation.

Reviewer #1: All comments have been addressed

Reviewer #2: (No Response)

2. Is the manuscript technically sound, and do the data support the conclusions?

Reviewer #1: (No Response)

Reviewer #2: Yes

3. Has the statistical analysis been performed appropriately and rigorously? 

Reviewer #1: (No Response)

Reviewer #2: Yes

4. Have the authors made all data underlying the findings in their manuscript fully available?

Reviewer #1: (No Response)

Reviewer #2: No

5. Is the manuscript presented in an intelligible fashion and written in standard English?

Reviewer #1: (No Response)

Reviewer #2: Yes

6. Review Comments to the Author

Reviewer #1: (No Response)

Reviewer #2: The Authors have answered most of my comments.

However, one of their answer to one of my previous comments is not clear to me.

“3. Did the Authors check whether the network measures were consistent across the range of thresholds explored?

No, we didn’t check. But probably the network measures were proportional to network density because of the weighted edges, as mentioned in M&M Functional network reconstruction.”

In the main text the Authors state that “Network metrics were calculated at different correlation matrix densities, from 20% to 50% density with a 5% interval, and averaged over these densities”. If they computed the metrics why they did not check for network measure consistency across the range of thresholds? They may easily check it by plotting network measure values against thresholds. While not emerging from the text that the network measures were proportional to network density because of the weighted edges, the fact that some of the network measures may be proportional to network density does not necessarily means that the topology of the network is stable across the thresholds.

In addition, could the Authors explain how they chose the specific density range 20-50%?

7. PLOS authors have the option to publish the peer review history of their article (what does this mean?). If published, this will include your full peer review and any attached files.

Reviewer #1: **Yes: **Marloes J.A.G. Henckens

Reviewer #2: No

---

## [Author Response · Author response to Decision Letter 2]

31 Jan 2023

Response to reviewers:

We would like to thank the reviewers again for their very helpful and detailed comments.

Reviewer #2: 

6. Review Comments to the Author

Reviewer #2: The Authors have answered most of my comments.

However, one of their answers to one of my previous comments is not clear to me.

“3. Did the Authors check whether the network measures were consistent across the range of thresholds explored?

In the main text the Authors state that “Network metrics were calculated at different correlation matrix densities, from 20% to 50% density with a 5% interval, and averaged over these densities”. If they computed the metrics why they did not check for network measure consistency across the range of thresholds? They may easily check it by plotting network measure values against thresholds. While not emerging from the text that the network measures were proportional to network density because of the weighted edges, the fact that some of the network measures may be proportional to network density does not necessarily means that the topology of the network is stable across the thresholds. In addition, could the Authors explain how they chose the specific density range 20-50%?

Thank you for this important comment. 

The reason why we chose to specify the density range to be 20-50% is because the typical range for densities is 5-50% (standard in GRETNA), and because it is a balance between including enough regions to maintain a detailed representation of the brain while excluding noise or irrelevant voxels that could interfere with the analysis. But it is not a fixed standard [1, 2]. In our study, we did not include the lowest densities because there were no connections at the lowest densities, as indicated in the manuscript line 209-212 “To remove the weakest connections, thresholds based on network density (i.e., the number of remaining connections divided by the maximum number of possible connections) were applied to the correlation matrices.” We chose to start from 20% since then it is possible to compute network measures for this density and wanted to keep as close to the approved range by GRETNA.

And we agree with the reviewer that some of the network measures may be proportional to network density does not necessarily mean that the topology of the network is stable across the thresholds. Thus, we did the check for the consistency of the network measures across the range of thresholds in amygdala (these are the network measures described in our article), the results were shown below. 

a

b

c

Fig. S1 The global efficiency (a), local efficiency (b), and characteristic path length (c) of amygdala in a range of sparsity thresholds (20%-50%, with 5% intervals)

As depicted in Fig.S1, even at low densities (20%), connections related to the amygdala are present in the surviving networks (5 regions) of 2 connections. This indicates that connectivity related to the amygdala is high and likely crucial. And starting at 35% densities, the results indicate a stable trend, which is convincing that the network measures are highly stable. In brief, the network measure consistency is stable across the range of 20% to 50% thresholds.

We added the consistency analysis results into the Supplement material and mentioned in the revised manuscript.

1. Rubinov M, Sporns O. Complex network measures of brain connectivity: Uses and interpretations. NeuroImage. 2010;52(3):1059-69. doi: https://doi.org/10.1016/j.neuroimage.2009.10.003.

2. Hallquist MN, Hillary FG. Graph theory approaches to functional network organization in brain disorders: A critique for a brave new small-world. Netw Neurosci. 2019;3(1):1-26. Epub 2019/02/23. doi: 10.1162/netn_a_00054. PubMed PMID: 30793071; PubMed Central PMCID: PMCPMC6326733.

---

## [Editor Report · Decision Letter 3]

8 Feb 2023

Network analysis reveals abnormal functional brain circuitry in anxious dogs

PONE-D-22-02320R3

Dear Dr. Xu,

We’re pleased to inform you that your manuscript has been judged scientifically suitable for publication and will be formally accepted for publication once it meets all outstanding technical requirements.

Kind regards,

Tamas Kozicz

Academic Editor

PLOS ONE
---

## [Editor Report · Acceptance letter]

10 Feb 2023

PONE-D-22-02320R3 

Network analysis reveals abnormal functional brain circuitry in anxious dogs 

Dear Dr. Xu:

I'm pleased to inform you that your manuscript has been deemed suitable for publication in PLOS ONE. Congratulations! Your manuscript is now with our production department. 

Kind regards, 

on behalf of

Dr. Tamas Kozicz 

Academic Editor

PLOS ONE